# Evaluating the Usability of mHealth Apps: An Evaluation Model Based on Task Analysis Methods and Eye Movement Data

**DOI:** 10.3390/healthcare12131310

**Published:** 2024-06-30

**Authors:** Yichun Shen, Shuyi Wang, Yuhan Shen, Shulian Tan, Yue Dong, Wei Qin, Yiwei Zhuang

**Affiliations:** 1School of Health Science and Engineering, University of Shanghai for Science and Technology, Shanghai 200093, China; 2The Institute of Rehabilitation Engineering and Technology, University of Shanghai for Science and Technology, Shanghai 200093, China

**Keywords:** usability evaluation model, eye tracking, entropy method, task analysis method

## Abstract

Advancements in information technology have facilitated the emergence of mHealth apps as crucial tools for health management and chronic disease prevention. This research work focuses on mHealth apps for the management of diabetes by patients on their own. Given that China has the highest number of diabetes patients in the world, with 141 million people and a prevalence rate of 12.8% (mentioned in the Global Overview of Diabetes), the development of a usability research methodology to assess and validate the user-friendliness of apps is necessary. This study describes a usability evaluation model that combines task analysis methods and eye movement data. A blood glucose recording application was designed to be evaluated. The evaluation was designed based on the model, and the feasibility of the model was demonstrated by comparing the usability of the blood glucose logging application before and after a prototype modification based on the improvement suggestions derived from the evaluation. Tests showed that an improvement plan based on error logs and post-task questionnaires for task analysis improves interaction usability by about 24%, in addition to an improvement plan based on eye movement data analysis for hotspot movement acceleration that improves information access usability by about 15%. The results demonstrate that this study presents a usability evaluation model for mHealth apps that enables the effective evaluation of the usability of mHealth apps.

## 1. Introduction

### 1.1. Background and Motivation

The widespread adoption of mHealth apps has resulted in time and cost savings for both healthcare providers and patients, while also improving the ease of monitoring and recording medical conditions [1,2,3]. Additionally, mHealth apps can be used as an intervention tool to monitor disease progression [4].

In the post-pandemic era, research indicates that there has been a sharp increase in the global demand for and usage of mHealth apps [5]. Several review studies and systematic surveys have demonstrated that the use of mHealth apps for health management is gaining popularity in China [6,7,8]. A significant portion of these data is attributed to mHealth apps for chronic disease management, which provide comprehensive, continuous, and proactive management services for patients with chronic conditions. Studies demonstrate the positive impact of mobile applications on chronic disease management, such as osteoarthritis, diabetes, and chronic pain [9,10,11]. A study has proposed a KneeOA app with features of behavioral change technology support, such as goal setting, action plans, and self-monitoring. And it has been validated that its interventions could be effectively accepted [12]. The design, development, and piloting of mHealth interventions for diabetes management has been widely validated, and the use of an app to help patients track their blood glucose levels, dietary intake, and exercise has been proved to be met with high levels of satisfaction from the patient population [13]. In the case of the pain field, medications are often insufficient to reduce pain. There is research on collecting pain data through the development of apps so that doctors can easily select patient-friendly treatments to reduce pain [14]. As the user base and market for mHealth apps continue to expand, designers must focus on improving user acceptance of their products [15]. Usability refers to the extent to which a product can be used by a specific user in a specific environment to achieve specific goals safely, efficiently, and comfortably. Evaluating the usability of mHealth apps is crucial for product development. A study of mHealth application development intentions suggests that current software usability fails to positively engage end users because of the lack of a unified regulatory regime proposing evaluation methods for usability [16].

As diabetes mellitus reaches epidemic levels worldwide, the need to explore effective medical measures for preventing and controlling its progression has become increasingly urgent. In recent years, China has emerged as one of the nations with the highest number of diabetic patients globally [17,18,19]. Compounding this issue, the crude and age-standardized mortality rates of diabetes in both urban and rural areas of China have shown significant increases [20], largely attributable to poor lifestyle choices, rapid urbanization, and other contributing factors. Several studies have co-authored the use of mHealth apps to demonstrate their feasibility in intervening in routine health management, particularly in the context of diabetes management in Chinese adults [21,22,23]. Although the use of mHealth apps holds considerable promise for effectively reducing the burden of diabetes, there is a relative lack of research on their practical application in China, as well as on user acceptance and willingness to use them.

Achieving management of diabetes is a widely discussed clinical issue, and although studies have shown that telemedicine services for glycemic control can be used to manage diabetes, patient adherence still needs to be improved to achieve good clinical outcomes. One study has suggested a relationship between user satisfaction and improvement in adherence or hemoglobin A1c (HbA1c), and it is believed that the enhancement of telemedicine services will be effective in improving clinical outcomes [24]. A study examining patient adherence and treatment effectiveness in a new mobile healthcare system (including mHealth apps and consultation platforms) in China also demonstrated that when patient adherence is improved, reductions in disease indicators (e.g., HbA1c) can be visualized clinically [25]. Similar studies have shown that by providing effective telemedicine services and self-management tools, patient engagement and responsibility for diabetes management can be increased, leading to improved treatment adherence and clinical outcomes [26]. With this in mind, the answer to the question of how to further improve adherence with the information-based healthcare model, which has been accepted by patients, has been focused on the issue of usability. One study reviewed currently offered glucose management applications and suggested that improving usability, perceived usefulness, and, ultimately, technology adoption are important ways to aid self-management [27]. 

The current usability studies that can be accessed suffer from several shortcomings: (1) They rely heavily on users’ subjective evaluations, which may lack objectivity or accuracy. Although there has been a surge in research activity in this area since 2013, most studies have focused on user satisfaction as an indicator, with the use of questionnaires being the predominant research method (68 of 96). Fewer studies have used performance indicators (25 of 96) and eye tracking (1 of 96). (2) They do not adequately consider the risks associated with the use of mHealth apps. Current usability research tends to focus on improving the user experience and operational efficiency of mHealth apps, but the risks and security issues associated with mHealth apps must be focused on due to their usage scenarios with specific user groups. Unlike everyday software, vulnerabilities, errors, or flaws in critical steps can lead to incorrect diagnosis, treatment, or medication administration. This can pose an immediate risk to patient health and safety. (3) The predominance of English-language interfaces in the studies may limit their cultural applicability. Cross-cultural adaptation has not been well focused on in the many studies presented. This mainly includes the localization efforts of the test software and the problem of direct translation of the test questionnaires. Achieving semantic and cultural equivalence when designing evaluations is a prerequisite for a reliable experimental structure [28,29,30].

### 1.2. Related Work

This chapter provides a review of existing studies and research related to the evaluation of usability in mHealth apps. The review encompasses various aspects of usability evaluation, including task analysis methods and the use of eye movement data.

#### 1.2.1. Usability Evaluation in mHealth Apps

Usability evaluation plays a crucial role in ensuring the usability of mHealth apps. Smith et al. conducted a systematic review of usability evaluation methods used in mHealth applications. The review highlighted common methods such as heuristic evaluation, user testing, and questionnaires, emphasizing the importance of these methods in identifying usability issues and improving the overall user experience [31]. Klasnja et al. proposed a framework that takes into account factors such as user engagement, ease of use, and user satisfaction in order to comprehensively assess the usability of mHealth applications. This study highlights the importance of adopting a rigorous evaluation methodology to facilitate the design of effective, user-centered mHealth apps, which will ultimately improve healthcare service delivery and patient engagement [32]. Based on this, the incorporation of more objective evaluation methods in the evaluation to enhance the reliability of the results is worthy of being looked at.

#### 1.2.2. Task Analysis Methods

Task analysis methods are vital for evaluating the usability of mHealth apps by understanding the cognitive processes and user interactions involved in task performance. Zayim et al. conducted a study that employed task analysis techniques to evaluate the usability of a mobile health app for the self-management of chronic diseases. This approach helped uncover usability challenges and inform interface redesign to improve task efficiency [33]. Additionally, Wildenbos et al. proposed a task analysis framework specifically tailored to assess the usability of mHealth apps for older adults. The framework combined cognitive task analysis (CTA) and observational methods to identify the cognitive processes, decision-making strategies, and user difficulties encountered by older adults while using an app [34]. 

These recent studies demonstrate the continued relevance and effectiveness of task analysis methods in evaluating the usability of mHealth apps. By analyzing the tasks and cognitive processes involved, these methods contribute to identifying usability issues and informing design improvements, ultimately enhancing the usability of mHealth apps.

#### 1.2.3. Eye Movement Data Analysis

Eye movement data analysis is a valuable approach used in the evaluation of usability in mHealth apps. Chamberlain provides a comprehensive review of eye-tracking techniques and their application in usability studies. By tracking users’ eye movements, researchers can gain insights into visual attention patterns, information processing, and interaction behaviors while using mHealth apps [35]. Asan et al. utilized eye tracking to evaluate the usability of a mobile app for medication management, analyzing users’ gaze patterns and fixations to assess the effectiveness and efficiency of task completion. This analysis revealed areas of the app’s design that required improvement to optimize usability. Incorporating eye movement data into usability evaluation provides objective and quantitative measures of user engagement, cognitive load, and visual attention, enabling researchers to make informed recommendations for enhancing the usability and interface design of mHealth apps [36].

In this work, we present an evaluation model based on task analysis methods and eye movement data. In contrast to previous studies, we evaluate the usability of a blood glucose self-management mHealth app by processing and analyzing objective data and suggest improvements to the application design. The feasibility of the evaluation model is verified by changing the usability metrics before and after design adjustments.

## 2. Materials and Methods

### 2.1. Usability Evaluation Model

Unlike previous research approaches, this work introduces a usability evaluation model for mHealth apps that combines traditional metrics with eye movement analysis. By integrating subjective and objective metrics, our model serves as a benchmark for enhancing mHealth apps and evaluating their messaging usability. The proposed usability evaluation model is shown in Figure 1.

Our model comprehensively evaluates the interaction usability of the test subject (mHealth app) across two dimensions: operational usability and information access usability.

We can divide usability into two main areas for designing experiments: operational usability and information access usability. Operational usability is concerned with the ease and efficiency with which users can actually operate a product or system. Ease of learning, ease of remembering, and efficiency of operation are key elements of operational usability. mHealth apps should be designed with intuitive and simple interfaces, with clear instructions and help files to help users get started and complete tasks with ease. Information accessibility is concerned with the ease and efficiency with which users can access the information they need. mHealth apps should provide a clear structure of information so that users can quickly find the information they need. By focusing on these two aspects of usability, one can assess whether a medical application has good usability. Within the operational usability aspect, we employed task analysis and an orientation questionnaire to assess the test subject’s performance. Task analysis offers design improvement suggestions and risk evaluation in terms of efficiency, potential errors, and heuristics by analyzing errors, time, and expressions of doubt during simulated task performance [37]. The orientation questionnaire collects user feedback on usability, interface design, and interaction satisfaction to prioritize improvements. We also utilized the System Usability Scale (SUS) questionnaire to validate the feasibility of the proposed improvements [38]. Appendix C shows the sus questionnaire we used.

The SUS score is calculated by subtracting the rating given by the subject from 5 for even-numbered statements and subtracting the rating given by the participant from 1 for odd-numbered statements. Then, all the scores are added together and multiplied by 2.5, and the final value is the SUS score. SUS scores range from 0 to 100, with higher scores indicating that the usability and satisfaction of the system are evaluated better by users. The higher the score, the better the user’s evaluation of the usability and satisfaction of the system [38]. In the information access usability aspect, we used eye tracking to measure the test subject’s performance [39]. The Tobil X1 Pro eye-tracking device tracks the user gaze during navigation and maps hotspots to provide insights into barriers to information access caused by suboptimal information display methods. We validated the feasibility of the proposed improvements by comparing user information acquisition scores before and after implementation. To verify the effectiveness of the proposed model, we conducted usability testing experiments on a high-fidelity prototype.

### 2.2. Experiment Design

As described in the model, the evaluation consists of two parts: operational usability evaluation and information access usability evaluation. In the operational usability evaluation, subjects complete a series of operational tasks according to a predefined task list and fill out the SUS questionnaire to assess their experience. We chose the SUS to measure user satisfaction because it is very effective in assessing user-perceived usability and is simple and economical [40]. Figure 2 depicts the experimental design for the operational usability evaluation.

In the information access usability evaluation, we recorded users’ eye movements as they navigated through three data presentations in the high-fidelity prototype [41]. Before viewing, the users were provided with questions related to content and completed the test by answering them. To assess the efficiency of user information acquisition, we selected four indicators (total fixation duration, time to first fixation, fixation count, and total visit duration) [42] and combined them with weight indices from entropy analysis to calculate a composite score for each user. Total fixation duration is the total amount of time for which the subjects gaze at a specific target and can be used to measure the attractiveness and importance of the target. Time to first fixation indicates the time when the subjects first gaze at a specific target after the start of the experiment, reflecting the attentional guidance and visual attractiveness of the target. Fixation count is the number of times the subjects gazed at a specific target during the experiment, reflecting the frequency of gaze and the level of interest in the target. And total visit duration is the cumulative time that the subjects visited the specific target in the experiment [43]. The prototype’s average information acquisition level was determined by calculating the mean of these scores. We employed the entropy weighting method to more objectively reflect the utility value of sample information entropy and derive more accurate indicator weights for evaluating user information access efficiency [44]. Figure 3 depicts the experimental design for the information access usability evaluation.

We conducted two rounds of experiments. The first round provided information on risk analysis, design satisfaction, and expert recommendations for prototype improvement. The second round was conducted on the improved prototype, and we compared the data from both rounds to assess the effectiveness of the proposed evaluation model in optimizing the interface based on differences in usability metrics. Ultimately, this comparison validated the usability of our evaluation model.

### 2.3. Subjects

For the first round of evaluation, 18 subjects were recruited for the test, and these 18 subjects were categorized into two age groups—User Group A and User Group B. The average age of User Group A was 21.67 years (Sd = 0.62 years), with 9 participants, while the average age of User Group B was 56.67 years (Sd = 2.42 years), with 9 participants. It is worth noting that the 9 people in User Group A had relatives with diabetes, while all members of Group B had diabetes. Table 1 shows the age distribution of the users in both groups. The purpose of this recruitment was to restore the ability of diabetic patients to perform self-management of blood glucose alone versus having a guardian record on their behalf due to factors such as vision, age, and mental ability [45]. 

For the second round of evaluation, a total of 16 subjects were recruited for this test, and these 16 subjects were divided into two age groups—User Group A and User Group B. The mean age of the members of User Group A was 21.92 years (sd = 0.79 years) for a total of 8 participants; the mean age of the members of User Group B was 55.75 years (sd = 5.56 years) for a total of 8 participants. In the second round of the evaluation, all subjects in Group A had relatives with diabetes, while all members of Group B were diabetic. Table 2 shows the age distribution of the two groups.

The subjects participating in both evaluations had watched the relevant operational video before the test and were left with a 24 h forgetting period to meet the real using environment. To ensure the accuracy of the evaluation results, repeated selections were avoided in the subject selection. The subjects selected for both experiments were as consistent as possible in terms of age, education level, and sample size. Notably, our subject selection was guided by the ISO 9241 human factors engineering standard (ISO 9241-210:2019 [46]).

### 2.4. Task and Procedure

#### 2.4.1. Prototyping

To avoid conflicts of interest, we designed a high-fidelity prototype for experimentation. The prototype was designed with a functional layout, with reference to the listed apps available in Apple’s app marketplace. The prototype offers four common functions: blood glucose recording, weight recording, record management, and insulin dose calculation. Figure 4 depicts the prototype’s home page, record page, and settings page. It is worth noting that during the evaluation test, we provided the Chinese interface to align with the cultural habits of the subjects. In order to demonstrate the functional layout of the interface, the translated images are shown here.

#### 2.4.2. User Notification and Pre-Training

Each subject signed an informed consent form and received pre-training on the test administration, and we were allowed to record the subjects’ behavior during the test for research purposes.

#### 2.4.3. Task Analysis and Error Recording

The first evaluation item was task analysis [47]. The test team consisted of a moderator, two recorders, and an equipment manager. During testing, the moderator issued tasks and responded to the users’ requests for operational assistance. The recorders documented the users’ usage errors, help requests, and verbal statements while performing tasks. The equipment manager ensured the proper operation of recording equipment, software, and the experimental platform.

Task analysis identifies design errors and predicts usage errors. We used the goals, operators, methods, and selection rules model (GMOS) to predict expected task completion time and calculated user task efficiency as the percentage deviation between actual and estimated completion times [48]. To ensure result accuracy, we designed eight task scenarios with a total of eight subtasks. Table 3 shows our designed task scenarios and subtasks.

#### 2.4.4. Eye Movement Tasks and Data Acquisition

After completing the task analysis, the subjects performed eye-tracking analysis of the prototype’s three data presentation forms in front of an eye-tracking device [49]. Figure 5 displays the three data presentation forms used in this evaluation, colors are used to visualize how dangerous the user’s blood glucose level is, where red indicates reaching a dangerous value, yellow warns of an imminent dangerous value, and green indicates safety. We surveyed blood glucose information display methods in similar mHealth apps on the iOS and Android markets and found that the most commonly used forms were line graphs, bar charts, and tables. The information access usability evaluation assessed user perception of these different presentation forms.

We used Tobil’s eye-tracking system, which includes a monitor, eye tracker, and eye tracker mainframe, for testing [50]. The subjects viewed displayed images in front of the eye tracker and completed tasks issued by the moderator. These tasks involved reading displayed information and included (1) finding the highest displayed blood glucose value, (2) finding the lowest displayed blood glucose value, and (3) finding the number of occurrences of high blood glucose segments. Figure 6 shows the eye-tracking system used in this study.

#### 2.4.5. SUS and Post-Task Questionnaire

At the end of the evaluation, each subject completed the System Usability Scale (SUS) and an orientation questionnaire to subjectively evaluate the prototype’s interactive usability. We used the SUS to obtain interaction usability and ease of learning scores for the prototype.

We designed the orientation questionnaire ourselves to collect post-use satisfaction data from the subjects [51]. The questionnaire assesses subjects’ user expectations for mHealth applications and their satisfaction with the provided prototype. This scale is in conjunction with a five-point Likert scale, which asks participants to rate their satisfaction with interface design, interaction mode, and functional usability.

## 3. Results

### 3.1. Risk Records and Use of Error Records Derived from the Task Analysis Method

During the usability test, two recorders meticulously documented the in-use errors committed by users while operating the prototype to ensure data completeness and accuracy. Record sheets are provided in Appendix B. Table 4 and Table 5 list some of the in-use errors for Groups A and B, and the number of times each error occurred. Subsequent analysis of the records showed that the younger subjects in Group A made the most errors when performing the two tasks of adding records and checking records, with 23 and 20 errors, respectively. In contrast, middle-aged subjects in Group B committed the most errors while adding records, with a total of 28 errors. For both groups, the fewest errors occurred during insulin calculation and the main user switching subtasks. These frequently occurring in-use errors will be prioritized for improvement in future prototype iterations.

The risk of in-use errors stems from design issues related to the interface’s style structure. To investigate the impact of the interface’s style structure on in-use errors, we categorized common mHealth app interface styles into four tiers: logical, display, interaction, and other. Table 6 presents our classification criteria. We analyzed the style structures implicated in each in-use error record, prioritizing those that occurred most frequently. Our analysis revealed that Group A subjects most frequently experienced in-use errors due to the UI logic navigation structure, UI interaction task structure, and UI display state structure. In contrast, Group B subjects were most likely to commit usage errors with the logic navigation structure, interaction feedback structure, and display state structure. These high-impact style structures will be prioritized for improvement in future prototype iterations.

As a component of the health service industry, mHealth apps play a crucial role in providing health management for individuals, including patient populations. The effectiveness of data interpretation significantly impacts user safety. We consider a subtask to meet critical task requirements if it satisfies any of the following criteria: (1) directly affects the user’s interpretation of blood glucose data (e.g., prompts for the current blood glucose unit), (2) indirectly affects the user’s interpretation of blood glucose data (e.g., the current main user status is unknown), or (3) impacts the accuracy of blood glucose data entry (e.g., clear expression of the time period). We have labeled mission-critical items in the in-use error records. These mission-critical items will be given the highest priority in future prototype improvements.

### 3.2. Comprehensive Satisfaction Scores Derived from the Orientation Questionnaire

At the end of the task, each subject was asked to fill in an orientation questionnaire (Appendix A) designed by us, which measured user satisfaction with the prototype in terms of interaction, design, and usability. We also compared the different criteria for the usability of the prototype between the A and B subject groups. To ensure data consistency, we used satisfaction percentages for the analysis. Figure 7, Figure 8 and Figure 9 show the different attitudes of the two groups of subjects in terms of interface interaction, interface design, and functional usability.

Group A had an overall positive attitude towards the interface design satisfaction dimension; the interaction mode satisfaction dimension had an overall positive attitude, with action efficiency having a large degree of dissatisfaction and time efficiency accounting for more than half of the neutral and negative attitudes; and the function satisfaction dimension had an overall significant positive attitude, with no negative attitudes. Meanwhile, Group B had an overall more positive attitude towards the interface design satisfaction dimension; the interaction mode satisfaction dimension had an overall neutral attitude; and the function satisfaction dimension also had an overall positive attitude, with only multi-user records showing a small amount of dissatisfaction.

In order to verify whether the data from this orientation questionnaire are statistically significant, we chose the ICC intra-group coefficient with Fisher’s exact test to verify the consistency and specificity of the data [52]. Table 7 shows the results of the statistical analysis. These two statistical indicators are recommended for use in small sample data.

The results show that all aspects are in the range of 0.8~1.0, with a strong degree of agreement, except for the interface interaction ICC coefficient values, which are in the range of 0.4~0.6, with a medium degree of agreement. For all three aspects, the *p*-values are less than 0.05, and the results are statistically significant. In Fisher’s exact test, the *p*-values are less than 0.05, thus demonstrating a specific difference between Group A and Group B.

The statistical data verified that there was a consistent pattern of questionnaire data for Groups A and B and that there was specificity in the differences between the groups, so there was an objective basis for our study of the questionnaire data. The data from the orientation questionnaire provide a basis for prioritizing usability improvements in terms of user satisfaction.

### 3.3. Eye Movement Acceleration

To assess the ease of information access when the subjects were presented with three different forms of information input, we measured eye movement acceleration and task completion time to track the participants’ information-seeking behavior. Figure 10 shows the mean eye movement hotspot maps of Group A and Group B collected during information access. The eye movement hotspot map shows the distribution of the subjects’ attention points in the map while completing the information acquisition task. In the test, the hotspot distribution of the two groups of subjects showed consistency. The largest number of hotspots was concentrated in the area where the data records were provided. To further identify information access issues in the interface design, we analyzed the eye movement acceleration data exported by Tobil Studio V1.0.4 using Matlab R2022b [53]. Figure 11 and Figure 12 show the hotspot movement acceleration curves for one subject in Group A and one in Group B, respectively.

Eye movement acceleration plays an important role in improving the usability of mobile applications. It is used to measure the speed and degree of change in the eye movements of a subject while using a mobile application. Eye movement acceleration provides information about the subject’s attention distribution and eye movement trajectory on the application interface, which in turn reveals the efficiency of the subject’s interaction with the application interface. Different types of eye movements, such as saccade, smooth pursuit, and fixation, have different acceleration characteristics. Research on the switching assistance of the subject’s eye movement type demonstrates its efficiency for information acquisition.

The results of the information access usability test showed that the subjects in Group A completed the task and reported to the facilitator faster than the subjects in Group B. Figure 13 illustrates the average time of completion for each group under each of the three types of charts. In the eye movement acceleration study, the subjects in Group A also always moved from the saccade period to the smooth pursuit period earlier.

During the test, we also recorded user confusion regarding data presentation in tabular format, further corroborating our hotspot movement acceleration calculations. The subjects who viewed the line graph form reflected a lack of clarity in the data labeling, but most of them were able to complete the task successfully and in the shortest time of the three forms. The subjects who viewed the bar graph form were also able to complete the task successfully, but found the red, yellow, and green colors in the graph to be confusing, which caused them to have to think before they could state the task. The subjects who viewed the table form took the longest time to complete the task, as they were initially unable to understand the meaning of the table.

Combining the above data, we can see that for young people in Group A or middle-aged people in Group B, line graphs and bar charts, which reflect trends, are more acceptable for everyday use. It is more efficient to use these two forms to convey information.

### 3.4. Prototype Improvement Checklist Design

In the interaction usability dimension, we evaluated in-use error records and created a priority matrix for prototype improvement by combining risk and critical task analyses. Figure 14 presents the priority matrix, with importance on the horizontal axis and satisfaction on the vertical axis. We placed structural styles targeted for improvement within the priority matrix.

In the benefit matrix, target improvement points are divided into four dimensions, with the red area representing the highest priority, the blue area representing the second-highest priority, and the green area representing the lowest priority. In this experiment, the UI logical structure style issue in adding records and the UI interaction structure style issue in checking records were assigned the highest priority. We modified the first-round prototype based on the priority matrix and structural style analysis of in-use errors, incorporating feedback from the subjects throughout the test. A total of 18 improvements were made to the high-fidelity prototype, following Nielsen’s Top 10 Usability Principles [54]. This article presents examples of improvements made to the add record and settings pages. Figure 14 illustrates some of these improvements before and after implementation.

We improved the record page to enhance system visibility, undo–redo functionality, and consistency. In the initial version (Figure 15a), the users had to click on three small icons on the page switch bar to add blood glucose values, test time periods, and test dates in sequence. However, our task analysis revealed that the icons were unclear and did not provide sufficient guidance for users to perform predefined actions. Consequently, we revised the Add Record page to resemble Figure 15b–d, where the page automatically advances to the next Add page after the users enter the required information. On each Add page, we included a prominent icon indicating the type of information required and designed a virtual keyboard and scrolling menu to add information that is displayed directly to visualize the Add form. Figure 16 shows three of our improved information presentation modules, Figure 16a–c is in three different forms.

### 3.5. Model Validation Data

#### 3.5.1. SUS Score

To evaluate the improvement in the interaction usability of our prototype derived from the proposed evaluation model, the subjects completed the SUS scale after both the first and second rounds of tests. The difference in the SUS scores was used to assess the feasibility of our model.

Table 8 presents the usability scores assigned by the subjects in the first round of tests. The difference between the usability and ease of learning scores was not significant for all subjects. However, the subjects in Group B assigned an average ease of learning score of only 43.75, indicating that our prototype performed poorly on this indicator for middle-aged individuals.

After improving our prototype, we conducted a second round of tests with a new group of subjects who did not participate in the first round. These subjects also completed the SUS questionnaire after finishing the simulation task. Table 9 presents the usability scores assigned by subjects in the second round of tests.

#### 3.5.2. Entropy Method Comprehensive Score Based on Eye Movement Indicators

To assess the impact of our proposed model on the usability dimension of information accessibility, we evaluated the feasibility of our model by comparing the comprehensive score of information accessibility based on the entropy weighting method of eye movement indicators before and after improvement.

We selected four eye movement indicators (total fixation duration, time to first fixation, fixation count, and total visit duration) and assigned weights to each using the entropy weighting method to calculate a comprehensive score. In this example, the four indicators are inversely related to information acquisition efficiency; smaller values indicate higher efficiency. These minimal indicators were normalized to obtain a standard matrix by calculating the difference between each value and the maximum value of its category. We determined the weight of each column vector in the criteria matrix by calculating its entropy and used these weights to compute the overall information access score for each subject group. Table 10 presents the performance of the line graph in terms of information acquisition by subjects in the first test.

After assigning weights to each indicator, we calculated the average indicator value for the subjects with different information presentation forms to compute their comprehensive information acquisition score. Table 11, Table 12 and Table 13 present a comparison of the information access scores from the two rounds of tests.

By comparing the entropy weights before and after the improvement, the score of the line graph form improved by 14.95%, the bar chart form improved by 15.16%, the table form improved by 9.16%, and the average score of the three forms improved by 13.33%. This improvement shows that the improvement for all three forms has been effective. Table 13 demonstrates the comprehensive scores obtained by Group A and Group B in each of the two rounds of tests. Both Group A and Group B received a boost in all three forms. Improvements to the information display problem led to greater progress in information access efficiency in Group B. In the line graph, Group B achieved an improvement of 23.79%. Meanwhile, in the bar graphs and tables, there is also an improvement of 14.83% and 10.88%. Group A, however, made a 15.5% improvement in the bar graph form, which is the largest improvement of the three forms.

## 4. Discussion

To verify the feasibility of our model, we validated it in two dimensions: interaction usability and information access usability.

In the interaction usability aspect, after improvement, we observed an overall increase in the SUS scores for all subjects. The average overall score improved by approximately 24%, with groups A and B showing increases of approximately 21% and 27%, respectively. This indicates that our improvements had a greater impact on the satisfaction of middle-aged users. In terms of usability and ease of learning scores, the middle-aged subjects in Group B experienced the most significant improvement in ease of learning scores, at around 50%. The difference in the SUS scores validates that our evaluation model-derived improvements resulted in a user-perceived improvement in usability for the test subjects, demonstrating the feasibility of our proposed model.

In the information access usability aspect, when the design issues identified through our evaluation model that affect the usability of information access were addressed, the overall score for information access improved for all three different information output formats by an average of 13%. The proposed usability improvements were most significant for line graphs and bar charts, with increases of 14% and 15%, respectively. The next most significant improvement was in the form of a line chart, with an increase of around 9%. Combining SUS and eye movement entropy weight data, we demonstrate the feasibility of our evaluation model for usability testing and usability enhancement of the mHealth app.

In our research, the differences between the youth group and the middle-aged group regarding interface needs were also discussed. In the operational tasks, there was no significant difference in user errors between the middle-aged and young groups. When talking about the design requirements, the middle-aged group and the young group had different concerns; the young group was more concerned with the efficiency of the behaviors resulting from the interface design; however, the middle-aged people were more concerned with the intuitive nature of the logic during the interaction. When it comes to the information access experiment, the execution time of the middle-aged group was longer than that of the young group. Unclear design language, such as the use of colors to indicate whether a blood glucose level is safe or not, the use of dotted lines to indicate the maximum and minimum values recorded over time, and unclear headers, made them more likely to be confused when accessing information. This is why the middle-aged group made more progress in accessing the information when we completed the modifications. Many articles on the usability of mHealth apps have demonstrated and validated the usability needs of different age or status groups, and have demonstrated that the background of the subject group has a significant impact on usability studies. As a group with a high prevalence of type 2 diabetes mellitus, the usability needs of middle-aged and elderly groups deserve more attention.

In this study, we demonstrate the feasibility of a usability model for evaluating mHealth apps. However, our study has several limitations: (1) our sample size is small, and future experiments should include larger samples to increase the accuracy of the model; (2) we tested a high-fidelity prototype, and future experiments should be conducted on mHealth applications that are already available on the market to better reflect real-world usage environments; and (3) there are limitations in the age distribution of our subjects, and future studies should include a more diverse sample to improve generalizability.

## 5. Conclusions

We propose a usability evaluation model for mHealth apps that combines subjective and objective data metrics and recommends a risk-based approach to improving interaction and information access usability. Usability testing of a high-fidelity prototype demonstrated that the improvements suggested by our model increased interaction usability by approximately 24% and information access usability by approximately 15%, while also mitigating any identified risks. These results support the feasibility of the proposed model. However, future studies should expand the sample size and diversity of educational backgrounds to further validate the generalizability of our model.

## Figures and Tables

**Figure 1 healthcare-12-01310-f001:**
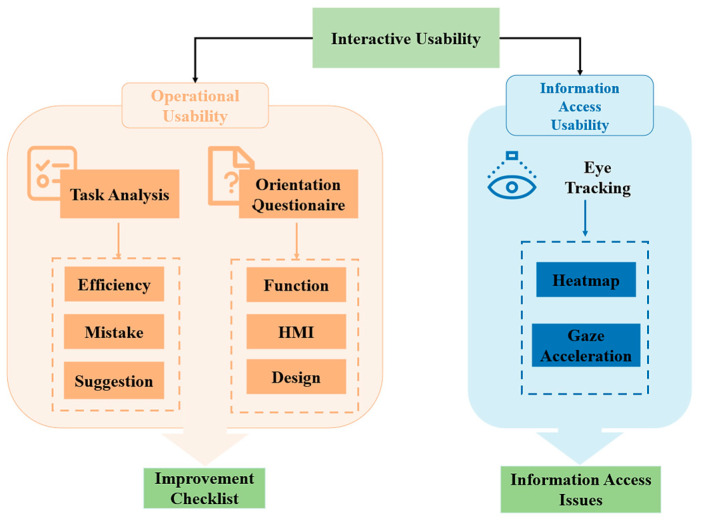
The usability evaluation model.

**Figure 2 healthcare-12-01310-f002:**
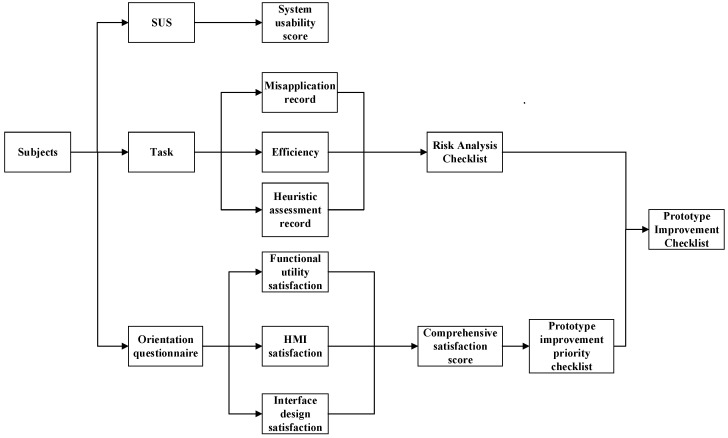
Experimental design of the operational usability evaluation.

**Figure 3 healthcare-12-01310-f003:**
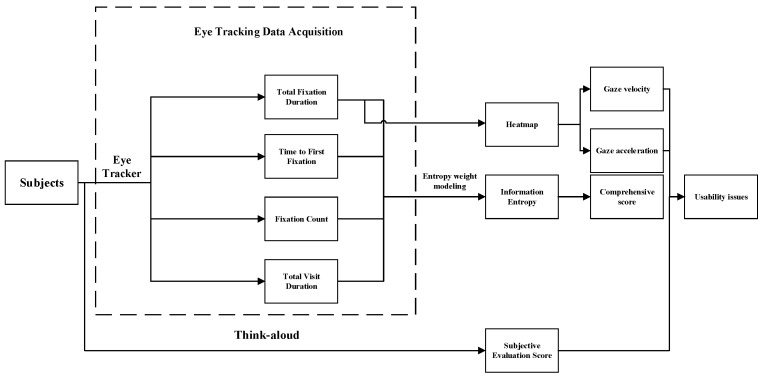
Experimental design of the information access usability evaluation.

**Figure 4 healthcare-12-01310-f004:**
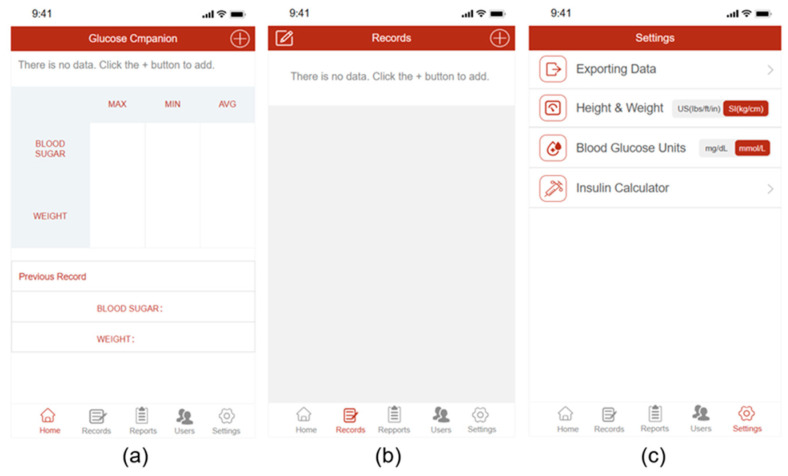
The prototype’s design: (**a**) the home page design; (**b**) the record page design; (**c**) the settings page design.

**Figure 5 healthcare-12-01310-f005:**
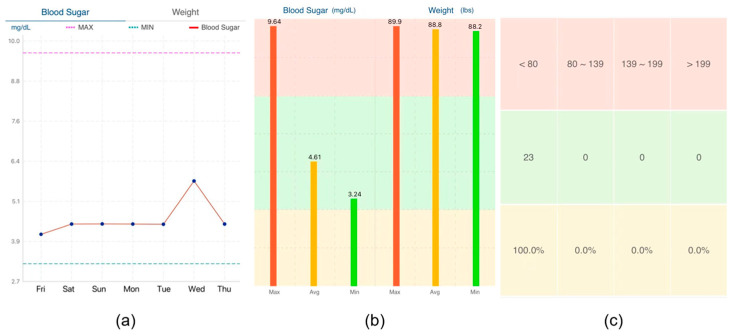
Three data presentation forms: (**a**) line chart form, (**b**) bar chart form, and (**c**) table form.

**Figure 6 healthcare-12-01310-f006:**
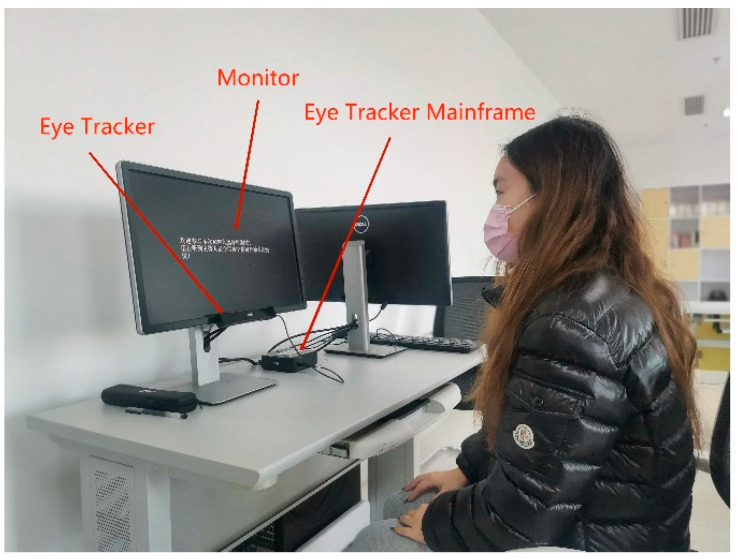
Eye-tracking experimental environment.

**Figure 7 healthcare-12-01310-f007:**
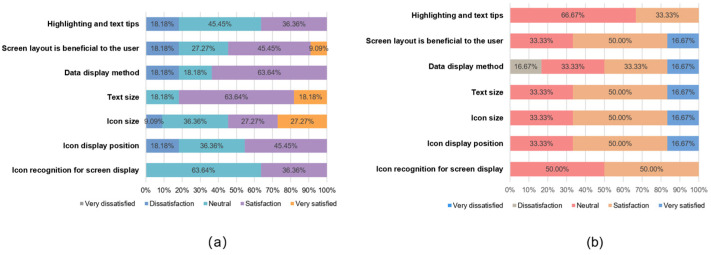
Interface design satisfaction comparison: (**a**) Group A; (**b**) Group B.

**Figure 8 healthcare-12-01310-f008:**
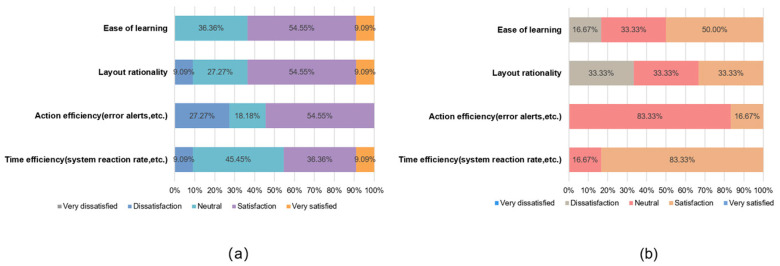
Interface design satisfaction comparison: (**a**) Group A; (**b**) Group B.

**Figure 9 healthcare-12-01310-f009:**
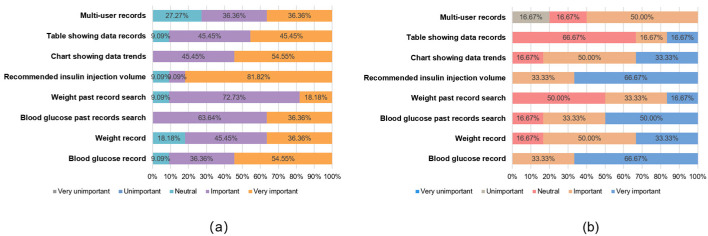
Functional and practical satisfaction comparison: (**a**) Group A; (**b**) Group B.

**Figure 10 healthcare-12-01310-f010:**
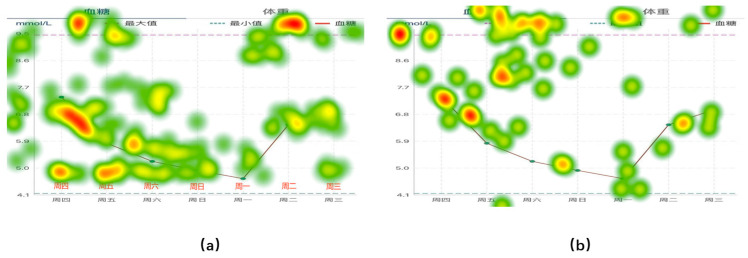
Mean eye movement hotspot maps of Group A and Group B. (**a**) Change curve of blood glucose level in one week for user Group A; (**b**) Change curve of blood glucose level in one week for user Group B.

**Figure 11 healthcare-12-01310-f011:**
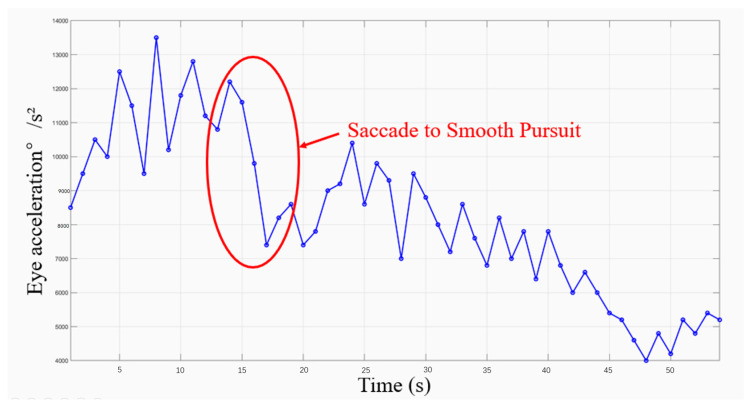
The eye movement acceleration of Group A.

**Figure 12 healthcare-12-01310-f012:**
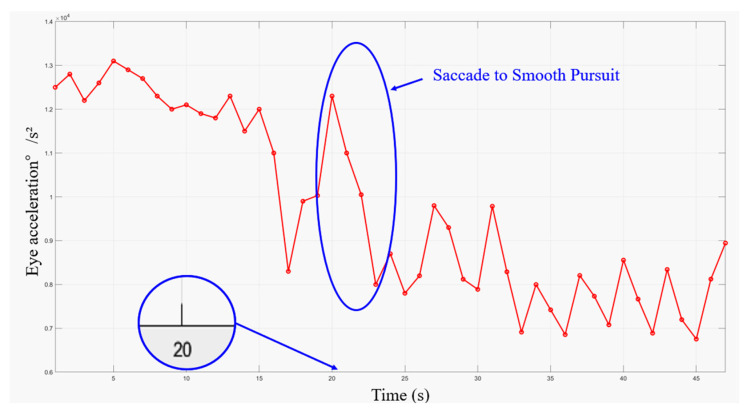
The eye movement acceleration of Group B.

**Figure 13 healthcare-12-01310-f013:**
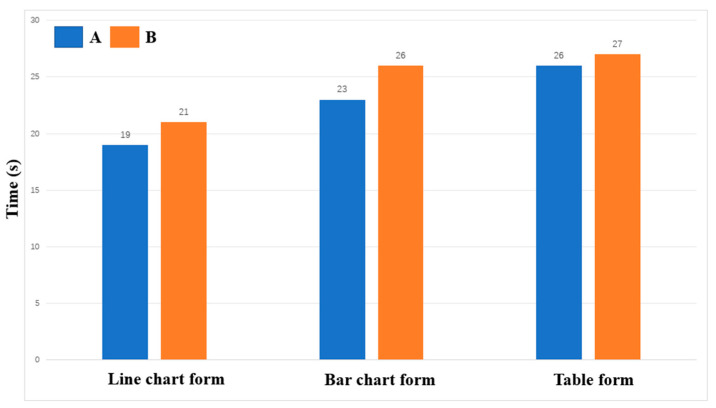
Average time of completion for each group.

**Figure 14 healthcare-12-01310-f014:**
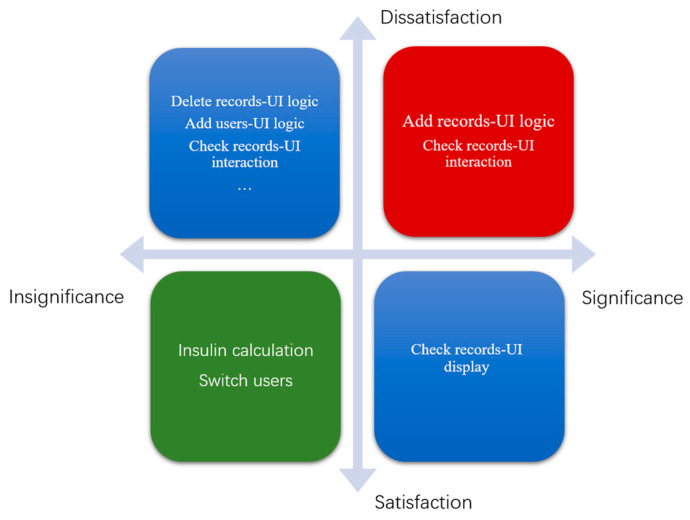
The priority matrix.

**Figure 15 healthcare-12-01310-f015:**
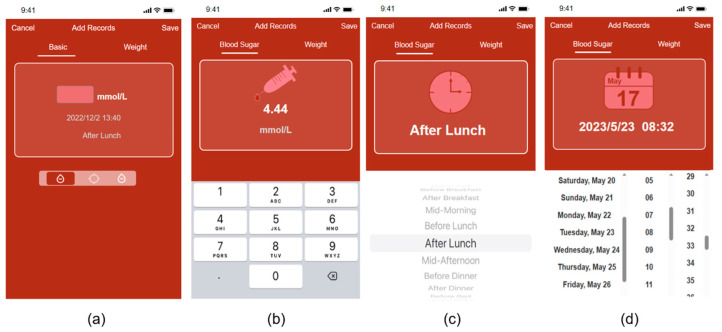
Comparison of before and after improvements to the Add Record page: (**a**) before improvements; (**b**–**d**) after improvements.

**Figure 16 healthcare-12-01310-f016:**
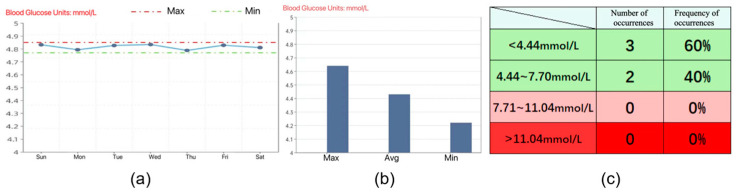
Three data presentation forms after Settings page improvements: (**a**) line chart form, (**b**) column chart form, and (**c**) table form.

**Table 1 healthcare-12-01310-t001:** Age distribution by group in the first round of evaluation.

Group	Ave	Sd
A	21.67	0.62
B	46.67	2.42

**Table 2 healthcare-12-01310-t002:** Age distribution by group in the second round of evaluation.

Group	Ave	Sd
A	21.92	0.79
B	45.75	5.56

**Table 3 healthcare-12-01310-t003:** Subtask design given to simulated new users.

Task scenario	New users manage blood glucose data for themselves	New users manage blood glucose data for others
Subtasks	Add user	Add user
	Add blood glucose records	Master user switching
	Check blood glucose records	Check blood glucose records
	Delete blood glucose records Insulin calculation	Add blood glucose records Insulin calculation

**Table 4 healthcare-12-01310-t004:** Experimental records for subjects in Group A.

Subtasks	Risk Description	Source of Risk	Risk Impact	Number of Occurrences	Is It Mission-Critical	Suggestions for Improvement
Add records	Blood glucose record added to the entrance is unknown	Interface navigation defects	Time-consuming increase in record addition	6	No	Add guidance tips
Add records	Time slot modification portal is not easy to find	Interface navigation defects	Modification time consumption increased	6	No	Add guidance tips
Check records	Blood glucose units are not visible enough	Interface display defects	Wrong perception of blood glucose amount may cause injury or death	6	Yes	Click on the blank form
Check records	Check the record time period modification method lack of consistency	Task interaction defects	Lower user satisfaction	9	No	Uniform time period modification method

**Table 5 healthcare-12-01310-t005:** Experimental records for subjects in Group B.

Subtasks	Risk Description	Source of Risk	Risk Impact	Number of Occurrences	Is It Mission-Critical	Suggestions for Improvement
Add records	The step to add blood glucose records is unknown	Interface navigation defects	Time-consuming increase in record addition	5	No	Add guidance tips
Add records	Ambiguous meaning of time zones	The meaning of the text is not clear	Unclear concept of blood glucose time recording	3	Yes	Adjust expressions in records
Add records	Time portal is difficult to find	Interface navigation defects	Interface navigation defects	6	No	Add guidance tips
Add records	Forgot to add weight information	Interface navigation defects	Easy to lead to imperfect information	3	No	Add guidance tips

**Table 6 healthcare-12-01310-t006:** The classification criteria of common styles.

Impact	First Tier	Second Tier	Third Tier
Interface interaction	UI logic	Menu	Main menu, sub-menu, menu tabs…
		Navigation	Main menu navigation, list navigation, search navigation…
		Icons	Static icons, dynamic icons
		Pop-up window	Notification pop-ups, warning pop-ups, type pop-ups…
Interface design	UI display	Menu interface	
		Status screen	Preview interface, multimedia content management interface, browsing interface…
		Function interface	Keying interface, search interface, photo interface…
		Other interface	Opening screen
Interface interaction	UI interaction	Interaction task	Confirm, enter, terminate…
		Interaction feedback	Send, save, delete…
Interface interaction Interface design	UI components	Interface area	Navigation bar, title area, content area…
		List type	Single selection list, multiple selection list, markable list…
		Operating components	Scrollbars, radio buttons, checkboxes…
		Text	Label name, column name…

**Table 7 healthcare-12-01310-t007:** The results of the statistical analysis.

DGP	ICC					Fisher’s Exact Test	
		Value	95% Confidence Interval		*p*	Value	Monte Carlo Significance
			Lower	Upper			
Interface design	Single	0.839	0.704	0.916	<0.01	41.365	<0.01
	AVE	0.913	0.827	0.956	<0.01		
Interaction mode	Single	0.494	0.064	0.766	0.014	60.792	<0.01
	AVE	0.661	0.121	0.867	0.014		
Functional practicability	Single	0.687	0.479	0.822	<0.01	31.296	0.019
	AVE	0.814	0.648	0.902	<0.01		

**Table 8 healthcare-12-01310-t008:** Overall SUS score for the first round of tests.

Item	SUS Score	Usability Score	Learning Score
All	57.79 ± 14.85	60.04 ± 17.21	58.09 ± 20.98
Group A	60.68 ± 16.45	59.38 ± 19.36	65.91 ± 17.75
Group B	52.50 ± 9.24	61.25 ± 12.25	43.75 ± 18.75

**Table 9 healthcare-12-01310-t009:** Overall SUS score for the second round of tests.

Item	SUS Score	Usability Score	Learning Score
All	71.67 ± 5.44	73.25 ± 4.76	67.35 ± 7.96
Group A	73.25 ± 7.85	71.50 ± 10.57	67.74 ± 13.40
Group B	66.75 ± 3.86	78.50 ± 5.58	66.03 ± 13.25

**Table 10 healthcare-12-01310-t010:** Entropy weights for access to information in the form of a line graph.

User	Time to First Fixation(s)	Total Fixation Duration(s)	Fixation Count(times)	Total Visit Duration(s)
U1	0.00	3.76	14	12.94
U2	0.00	1.95	7	3.37
U3	0.00	2.51	9	3.15
U4	3.02	1.71	10	3.58
U5	1.10	3.85	12	4.31
U6	0.00	2.74	11	3.90
U7	1.45	3.22	9	3.53
U8	2.16	3.12	11	3.98
U9	0.30	2.79	7	4.27
U10	0.00	3.72	8	7.85
U11	0.00	6.60	17	8.90
U12	0.00	6.32	13	5.33
U13	4.42	4.16	14	7.90
U14	0.01	3.65	13	5.25
U15	0.74	0.82	4	5.94
U16	0.00	4.40	13	3.92
U17	0.97	3.23	12	4.35
U18	0.00	3.70	11	3.58
Information entropy	0.97	0.95	0.95	0.97
Weight	0.20	0.30	0.32	0.18

**Table 11 healthcare-12-01310-t011:** Comprehensive information access score for the first round of tests.

Form	Time to First Fixation(s)	Total Fixation Duration(s)	Fixation Count(times)	Total Visit Duration(s)	Comprehensive Score
Line graph	0.79	3.45	10.83	5.33	5.62
Bar graph	0.83	3.44	10.65	4.91	4.75
Table	0.87	3.52	10.86	4.99	4.04
Ave	0.83	3.47	10.78	5.08	4.80

**Table 12 healthcare-12-01310-t012:** Comprehensive information access score for the second round of tests.

Form	Time to First Fixation(s)	Total Fixation Duration(s)	Fixation Count(times)	Total Visit Duration(s)	Comprehensive Score
Line graph	0.73	3.47	7.90	4.53	4.78
Bar graph	0.86	3.23	8.42	5.01	4.03
Table	0.79	3.19	10.73	4.77	3.67
Ave	0.79	3.30	9.01	4.77	4.16

**Table 13 healthcare-12-01310-t013:** Group A and B score comparison.

Form	Group A First Round	Group A Second Round	Group A First Round	Group A Second Round
Line graph	5.23	4.98	6.01	4.58
Bar graph	4.91	4.15	4.59	3.91
Table	3.76	3.49	4.32	3.85
Ave	4.63	4.21	4.97	4.11

## Data Availability

Data are contained within the article.

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
