# Peer review of "Evaluating the Usability of mHealth Apps: An Evaluation Model Based on Task Analysis Methods and Eye Movement Data"

_healthcare, 2024, doi:10.3390/healthcare12131310_

Round 1

Reviewer 1 Report

Comments and Suggestions for Authors

In this paper, the authors propose a usability evaluation model for mHealth apps.

Suggestions and questions (answers may/should be used to improve the manuscript):

1. The title is "Evaluating the usability of mHealth app: An evaluation model based on task analysis and eye movement data"; but what task? it could be defined in the title, or it could be in the plural form (i.e., "task analysis methods").

2. This sentence is big and confusing "This study describes a usability evaluation model that combines task analysis and eye movement data, while a blood glucose logging application was developed to validate the feasibility of the model by testing the usability of the program using the model."; What 'program'? Did you develop two proposals to evaluate each other?

3. Consider "The results demonstrate that this study presents a generalized usability evaluation model for mHealth apps that enables a comprehensive evaluation of the usability of mHealth apps."; this text is redundant and confusing. what is 'generalized usability' and 'comprehensive evaluation of the usability'? Why 'results demonstrate that this study presents'? Are the results from the study? too confusing.

4. Introduction. Usability and user experience are different. This is not clear in lines 42-47.

5. In lines 60-64, unpack (i.e., explain in detail) the 'shortcomings'.

6. Consider 'Unlike previous research approaches...' - line 72; what are the previous research approaches? What is the novelty of the study when compared to the previous works? A description of related work is completely missing. A new section called 'Related Work' would improve the quality of the manuscript.

7. Figure 1 - Orinetation -> Orientation. Improve the resolution of the image.

8. Avoid short paragraphs such as lines 79-81, 96-97, etc.

9. Figure 1 is not presented in detail. What is 'operational usability dimension'? Why 'operational'? Why 'dimension'? Explain them. Consider 'We also utilized the System Usability Scale (SUS) questionnaire to validate the feasibility of proposed improvements. [19]'; What improvements?

10. Also, consider 'To verify the effectiveness of our proposed model, we conducted simulated usability testing experiments on a high-fidelity prototype.'; what are 'simulated usability testing experiments'? high-fidelity prototype of what?

11. 'Evaluation' and 'Assessment' are words used interchangeably in the manuscript. In the context of the manuscript, are they the same?

12. What is 'interactive' usability evaluation (figure 2)? Why 'interactive'? The text has "Figure 2 depicts the experimental design for the 'operational' usability 'assessment'."

13. Explain "Total Fixation Duration, Time to First Fixation, Fixation Count, Total Visit Duration".

14. I could not understand "It is worth noting that in the evaluation test, we will present the Chinese interface to match the cultural habits of the subjects."; 'Will'?

15. The abbreviation SUS is multiple times defined in the manuscript. Only the first time is required.

16. Detail SUS and how its score is calculated.

17. What does "..." mean in table 6 - 'Third tier' column?

18. Orientation questionnaire designed by the authors must be provided as supplementary material or appendix. I am not sure what is presented in Appendix A and Appendix B. Provide headings to them.

19. Figure 10 is not in English and hard to understand. What do the X and Y axes represent? What is "mmol/L" in that figure?

20. Interpret Figures 11 and 12. What do they mean? Explain them.

21. Tables 10 -> Table (singular)

22. What can be concluded from the results presented in section 3.5.2.?

23. Content presented in lines 396-397 is repetitive.

24. Discussion section is poor. Some questions could be answered in that section:

- Are the findings similar to other studies?

- What can be concluded from the results?

- What are the limitations of the study?

- What are the future works?

25. What makes this study specific to 'mHealth apps'?

Reviewer 2 Report

Comments and Suggestions for Authors

Introduction

'The widespread adoption of mHealth apps has resulted in time and cost savings for 29 both healthcare providers and patients, while also improving the ease of monitoring and 30 recording medical conditions.' Needs referencing 

'According to the 2023-2028 China 34 Mobile Healthcare Industry Investment Planning and Forecasting Report, users are expected to continue to grow.' Needs referencing 

' Studies demonstrate the 38 positive impact of mobile applications on chronic disease management such as osteoarthritis, diabetes and chronic pain.' Please give more detail 

The introduction presents a good overview of mHealth in diabetes however it lacks any real detail in regards to why the evaluation model is needed and what it will add to the literature. 

Methods 

Please explain how the usability evaluation model was developed, figure 1. 

'Within the operational usability dimension, we employed task analysis and an orientation questionnaire to assess the test subject’s performance.' Please explain how this was done. 

Results and Discussion

I have not reviewed the results and discussion section. As the introduction and results need significant editing and after this I would suggest the authors review the results and discussion and edit to reflect the changes made earlier in the manuscript.  

In short, I think this could be a very interesting paper. However the authors do not present solid reason in regards to why this research is needed. The methods also contain a lot of waffle.  

Round 2

Reviewer 1 Report

Comments and Suggestions for Authors

The authors answered all my questions, and addressed my concerns.